# A Hierarchical Design Framework for the Design of Soft Robots

**Philip Frederik Ligthart *** [ID]**, Martin Philip Venter** [ID]

Department Mechanical and Mechatronic Engineering, Stellenbosch University, Stellenbosch 7600, South Africa
* Correspondence: 21031193@sun.ac.za

**Abstract:** This paper demonstrates the effectiveness of a hierarchical design framework in developing environment-specific behaviour for fluid-actuated soft robots. Our proposed framework employs multi-step optimisation and reduced-order modelling to reduce the computational expense associated with simulating non-linear materials used in the design process. Specifically, our framework requires the designer to make high-level decisions to simplify the optimisations, targeting simple objectives in earlier steps and more complex objectives in later steps. We present a case study, where our proposed framework is compared to a conventional direct design approach for a simple 2D design. A soft pneumatic bending actuator was designed that is able to perform asymmetrical motion when actuated cyclically. Our results show that the hierarchical framework can find almost 2.5 times better solutions in less than 3% of the time when compared to a direct design approach.

**Keywords:** optimisation; finite element; reduced-order-modelling; soft robot; hierarchical design framework; soft actuator; pneumatic network

## 1. Introduction

Soft robots have a design space that is complex and co-dependent. As a result of the highly compliant materials, successful designs require consideration of both shape and behaviour [1]. Soft robots are more biologically compatible than traditional robots, with potential applications in fields such as human-to-machine interfaces, search and rescue, and other delicate operations [2]. Despite the complex shape-dependant control, high degree of deformation, and non-linear material behaviour characteristic, it is still the convention to use a build-and-test development strategy [3]. However, the traditional build-and-test design paradigm is expensive and does not allow rapid exploration of the design space. Attempts have been made to use computational resources to aid in the design process, but these also have drawbacks, such as high computational expense. Researchers have made recent advancements in modelling [4,5] and optimising [6,7] techniques for soft robots that improve accuracy and computational efficiency [8]. Nonetheless, these methods often exist in isolation and provide limited benefits for designing a full robot [9]. For this reason, Pinskier and Howard [10] propose a hierarchical design framework to make the design process more efficient and allow for future modelling techniques' integration. In this paper, we demonstrate a hierarchical design framework to enable the efficient design of soft robots. Using a hierarchical framework, we aim to improve the design process and make it easier to incorporate new modelling methods.

## 2. Background

Soft robotics is a promising field of robotics that offers unique benefits not achievable with traditional rigid robots. Soft robots possess both active and passive compliance due to their complex material properties, which are costly and difficult to achieve with rigid robots [11]. This compliance renders them safer for human–machine interactions and ideal for delicate handling operations such as fruit picking [2,12]. Additionally, their material properties allow them to deform and adapt to their environment, making them suitable for navigating unstructured terrains. Hence, they hold great potential for search-and-rescue

missions [13]. The advent of 3D printing technology has enabled the complete manufacture of soft robots, including the incorporation of sensory equipment within their morphology, allowing for embodied intelligence and control [14,15]. Given these advantages, soft robotics is poised to have a significant impact on the future of robotics.

In the field of soft robots pneumatic actuation is the dominant actuation technology [16]. For this reason, we focus on pneumatic networks (PNs) in this paper. There are two main types of PNs, namely the slow pneumatic network (sPN) [17] and the fast pneumatic network (fPN) [18]. Many soft robots which have been developed are created by assembling multiple of these actuators together. Marchese et al. [19] created a soft robotic fish tail capable of escape manoeuvres using two sPNs, Schiller et al. [20] designed a gecko-inspired soft robot using six fPNs connected together, and Ilievski et al. [17] created a soft robotic gripper using six connected sPNs. In many of the designs found in the literature, a trial-and-error design approach is followed [3]. Both physical testing and FE simulations are used during the trial-and-error approach. This direct design process is feasible when the problem is simple and small; however, when a more elaborate problem is to be solved, it is difficult to manually iterate a design to achieve an acceptable solution, so to this end, authors have used computational optimisation techniques, such as gradient-based optimisation or genetic algorithms for the iteration of the designs [6,7,21].

Using computational resources to optimise the design process is advantageous for the designer, as it alleviates the need for manual iterations. Frameworks, such as those developed by Runge et al. [22], have been developed which incorporate both FE simulations and kinematic modelling techniques to allow for the automated design of soft robotics. However, such frameworks remain largely parametric and hence are limited in terms of design space exploration. Evolutionary design methodologies have also been explored with good success [10,21,23]. Many works make use of simulated environments to evolve the soft bodies; this is pragmatic but leads to a simulation-to-reality gap [10]. Other design frameworks have been developed that use topology optimisation and evolutionary methods to automatically design soft actuators and components [6,7]; however, when considering material nonlinearities, these methods are also plagued with a high computational burden. For an in-depth review of the autonomous design of soft robots, the reader is referred to Pinskier and Howard [10].

## 3. Materials and Methods

The hierarchical framework described here builds on the work of Pinskier and Howard [10]. The hierarchical framework presented here is aimed at condensing the large design problem into smaller sub-problems that are simpler or computationally more efficient to solve. The smaller sub-problems can then be assembled back into the large design problem. In this approach, the designer makes high-level decisions, and computational resources are utilised to optimise smaller sub-problems.

The hierarchical framework is comprised of multiple levels, shown in Figure 1. Level 1 constitutes the desired outcome of the entire design process. High-level decisions are made to simplify the design. The simplification of level 1 becomes level 2. The same holds when moving from level 2 to level 3. Simplifications can comprise splitting the design into sub-components, making use of symmetry, and repetition, among others. Each simplification is represented as a hierarchical step (HS) within the level. There can be multiple HSs in a single level, as in the case where a desired robot is split into multiple sub-components. For example, a rehabilitation glove can potentially be split into five sub-components, each representing a different finger. Once the design has been sufficiently reduced or simplified in the forward pass of the framework, each HS is then solved, starting from the last level. Each next HS that is solved is assembled back into the complete design in the first level. This is the backward pass through the framework.

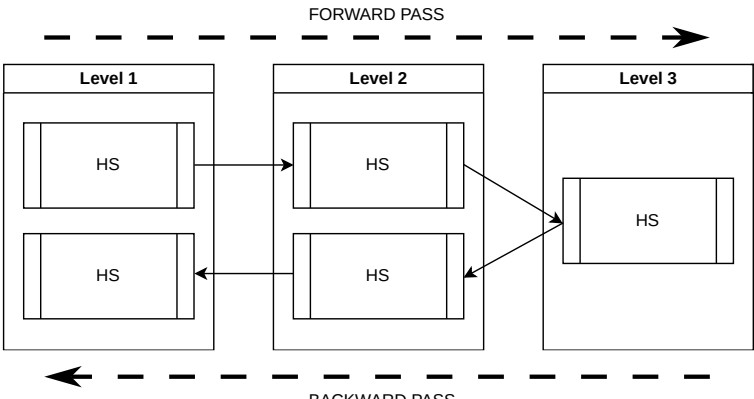

**Figure 1.** Hierarchical framework made up of multiple hierarchical steps. The complete design is broken down into simpler design problems in the forward pass. The simpler design problems are then solved and reassembled in the backward pass. The directions of the forward and backward passes are represented by the dashed arrows.

Within the framework, each HS is treated as a separate design problem. A single HS is shown in Figure 2. It must have a clearly defined target, properties, and constraints that are enforced by the designer based on application specifications and availability. These constraints can also help to reduce the design space. The design problem can often be framed as an optimisation problem, but it is important to make sure the optimisation is well defined. Optimisations can lead to unexpected or undesirable outcomes if the target properties or constraints are inaccurate as noted by Lehman et al. [24]. Clearly defining properties at each level of the hierarchy helps keep the design pragmatic and avoids unnecessary optimisation. It is important to remember that each HS in the design framework has its own distinct target; this is opposed to a single optimisation step, where a single objective function is to be achieved. When reducing the large problem into smaller sub-problems, the designer must carefully consider the effects of the simplifications. Using repeating units or incorporating symmetry can have consequences that restrict the design space.

Within the design framework, FE simulations can be utilised. However, when the FE simulation requires a non-linear analysis, the computational demands can be substantial. To address this challenge, reduced-order models can be employed to enhance the efficiency of optimising sub-problems. Many such models can be found in the literature, for example, piecewise constant curvature [4], Cosserat rods [5], topology optimisation [7], or a bespoke reduced model specifically tailored for the application at hand can be developed.

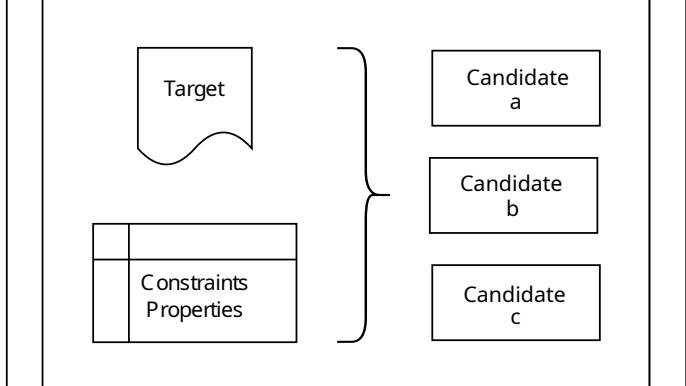

**Figure 2.** Single hierarchical step representing an optimisation problem. Multiple candidate solutions can be generated, which may be used in succeeding HSs.

## 4. Design of Asymmetrical Bending Motion Actuator

To demonstrate the above-described framework, a soft pneumatic actuator capable of performing asymmetrical bending motion is designed. Asymmetric motion is the objective for two reasons. Firstly, asymmetrical motion is essential for soft robots that locomote or swim. During forward motion, a walking or swimming robot requires its legs to follow an asymmetrical path. Furthermore, such an actuator presents a more complex design than a conventional soft pneumatic bending actuator (SPBA). Figure 3 provides an overview of the hierarchical framework implemented in the case study. The framework starts with the desired design in the first level, progressively breaking it down into simpler problems. The optimisation commences from the last level, working back to the first level. A summary of each level's function is presented in Table 1 and is discussed below.

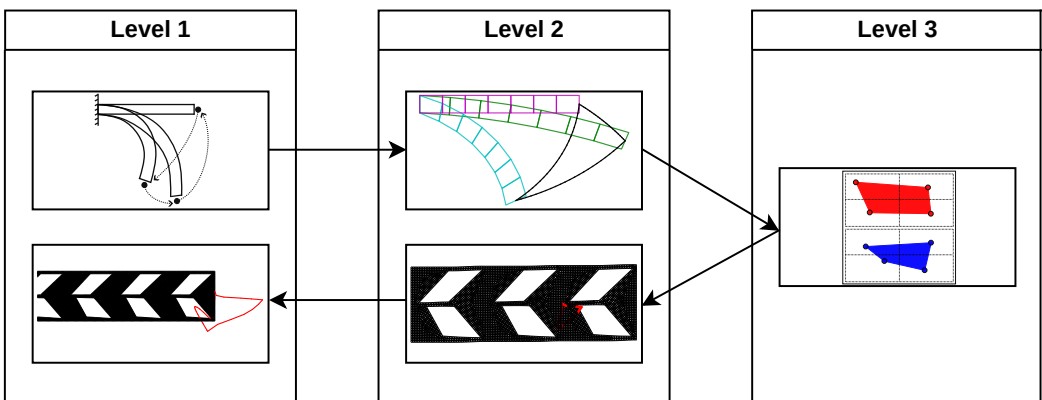

**Figure 3.** A complete hierarchical framework with figure representations of each hierarchical step in the framework.

**Table 1.** Hierarchical case study summary.

|  | Level 1 | Level 2 | Level 3 |
| --- | --- | --- | --- |
| Forward Pass | Desired design | Create reduced model | Chamber design |
| Backward Pass | Cascade to full length | Optimise 3 element model | Chamber design |

### 4.1. Forward Pass

When transitioning from the first level to the second level, a simplification is made by using repeating elements instead of a single continuous morphology. This simplification limits the possible deformations, but it is deemed acceptable for the desired outcome. A custom, simplified model was created to represent the actuator and to optimise the problem. The model is a $2 \times 2$ transformation of points, representing the movement of an element being actuated. Despite the non-linear path followed by a soft actuator, the start and end points of each actuation are only considered at this point, allowing for the use of a linear transformation. This simplification represents the model well enough to allow for rapid exploration with a low computation expense. While this approach represents a simplification of the actual model, it is permissible since later stages in the framework allow us to fully describe the model and perform more complete optimisations [25]. When the transformation is applied to a point (vector), it moves in space. Figure 4a shows an example of the two right-hand nodes of a unit being transformed. Figure 4b shows the result of when the two right-hand side nodes of each element in a cascade of connected elements are transformed. The figure shows that the model provides a good representation of an sPN during actuation. Mathematically, a point $[x_0, y_0]^\top$ is transformed into the point $[x_t, y_t]^\top$ using the equation

$$\begin{bmatrix} x_t \\ y_t \end{bmatrix} = \underbrace{\begin{bmatrix} x_{scl} & x_{sh} \\ y_{sh} & y_{scl} \end{bmatrix}}_{\text{Transformation}} \begin{bmatrix} x_0 \\ y_0 \end{bmatrix} \tag{1}$$

where *scl* represents scale and *sh* represents shear in the respective coordinate directions.

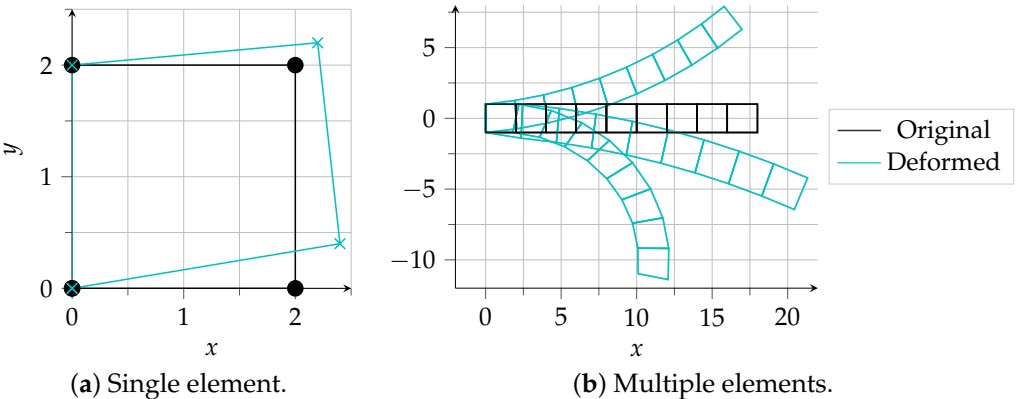

(**a**) Single element.  (**b**) Multiple elements.

**Figure 4.** Custom reduced order representation of an sPN undergoing actuation.

We used the reduced model to explore the design space. When examining the path traced by the actuator's tip, we observed that if there is only a starting point and one actuation point, the resulting path would be a line connecting the start and actuation point, i.e., start–actuation point–start. To create a closed area, at least one additional point is required. Thus, it was found that asymmetrical motion can be achieved by applying two consecutive transformations followed by a return transformation to the original state. Specifically, the path would be, start–actuation point 1–actuation point 2–start. These findings show that asymmetrical motion is possible, and to achieve it, the actuator will require at least two discrete actuations. Using the knowledge gained from level 2, a single unit will be designed that has two discrete chambers within it. The two chambers will be actuated independently. The properties and constraints for the forward pass are shown in Table 2.

**Table 2.** Hierarchical framework forward pass properties. Properties were chosen based on similar studies. Material [16], Geometry [6,26,27], Pressure [6,18,27].

| Property | Value/Specification |
| --- | --- |
| Material | Mold star 30 |
| Pressure | 0 to 15 kPa |
| Internal Chambers | 2 |
| Unit dimensions | $80 \times 80 \times 2$ mm |
| Number of units | 9 |
| Minimum sidewall thickness | 2 mm |

### 4.2. Backward Pass

The backward pass solves the simplified problems generated in the forward pass. This starts with the chamber layout optimisation in level 3. The chosen chamber layout is to have two independent chambers placed above and below one another within the repeating element. Two chambers were used to allow for two distinct actuations. This is graphically represented in Figure 5a. The two chambers are actuated according to Figure 5b. The *x* and *y* coordinates of each chamber corner node point are the design variables which are optimised. The design variables are represented as

$$\text{x} = [x_1, y_1, x_2, y_2, x_3, y_3, x_4, y_4, x_5, y_5, x_6, y_6, x_7, y_7, x_8, y_8] \tag{2}$$

The constraints from Table 2 were added to the optimiser as well as additional bounds to each node of the chambers to enforce the minimum wall thickness constraint, as seen in Figure 5a. An objective function is constructed where asymmetrical bending motion is desired. The asymmetry is quantified as the area enclosed by the tip of the actuator during a complete actuation cycle. The coordinates of the actuator tip are extracted during actuation, and the surveyor's area formula [28] is used to calculate the enclosed area. The formula makes use of the Cartesian coordinates of each vertex of a simple polygon. The formula is

$$A = \frac{1}{2} \sum_{i=1}^{n} (y_i + y_{i+1})(x_i - x_{i+1}) \tag{3}$$

As the optimisation is a minimisation problem, the objective function is the negative of the calculated area $A$.

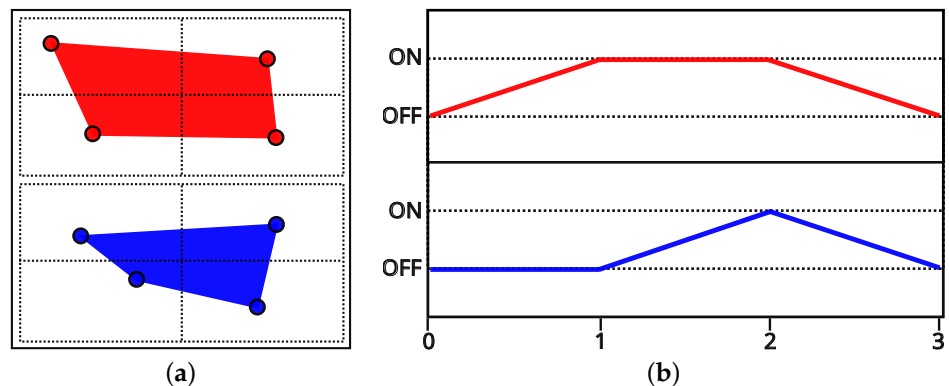

**(a)**　　　　　　　　　　**(b)**

**Figure 5.** Unit definitions. (**a**) The chamber layout within a unit. The dotted lines represent the boundaries of each chamber node. (**b**) Actuation pattern. Both chambers start and end at the off position, and this allows for cyclic actuation. Each coloured line corresponds to the actuation of the matching coloured chamber.

An additional simplification of reducing the number of elements in the model was made in order to reduce the computational burden of simulating the actuator. A collection of three units is the smallest possible representation that describes the actuator. When the repeating unit is cascaded to form the full-length actuator, it is in effect only the second element that is repeated, as the first and last elements have unique boundary conditions [29]. This is more clearly represented in Figure 6.

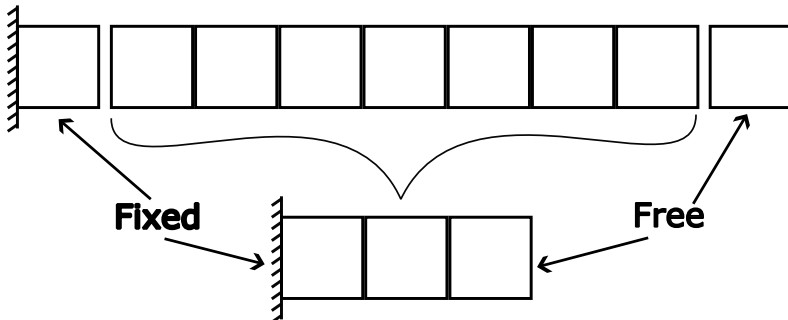

**Figure 6.** Reduction of a full-length actuator to its simplest representation. All units with unique boundary conditions remain in the reduced representation. Units with similar boundary conditions are condensed into a single unit representative of the group [29].

A non-linear finite element solver was used to simulate the units when the pressures were applied to the internal voids according to Figure 5b. The optimiser was then used to minimise the objective function. An optimisation was initialised using random parameters,

and multiple restarts were used. The properties of the non-linear solver and the optimiser can be found in Appendix A.

　　Figure 7 shows an optimised three-element actuator. The red line shows the path followed by the tip of the actuator during actuation. When the optimised element is cascaded to the full nine elements, the resulting actuation is shown in Figure 8. The figures show that the goal of achieving asymmetrical motion is achieved.

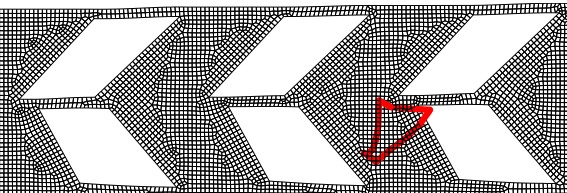

**Figure 7.** Three-unit finite element result showing the optimised chamber designs. The red line shows the enclosed area, which was maximised during the optimisation. For the three-element method, the midpoint of the right-hand side of the middle element is used as the tip to avoid the bulging of the tip having too large of an impact on the optimisation.

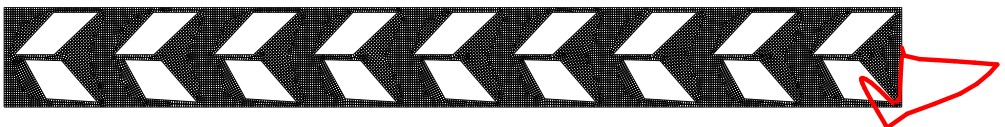

**Figure 8.** Result from the three-element optimisation design expanded to the full nine-unit actuator. FE mesh is shown in black with the tip displacements traced in red.

### 4.3. Results and Discussion

　　To evaluate the effectiveness of the hierarchical design framework, or hierarchical framework method, we also used a direct design approach as a comparison. The direct method was not designed using the hierarchical framework. Instead, it used all the nodes of all the chambers as design variables, and the entire actuator was simulated. This approach is referred to as the direct method. Figure 9 illustrates the graphical representation of the two different methods.

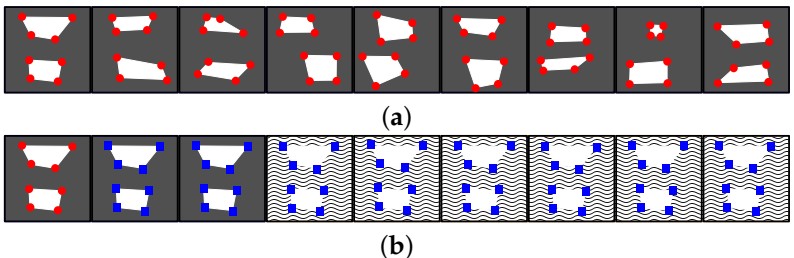

(**a**)

(**b**)

**Figure 9.** The different design method representations. Red dots represent design points, blue squares represent repeated points, the grey background shows FE simulated, and the wavy background shows non-FE simulated. (**a**) A complete domain where every chamber node is a design point and the entire actuator is simulated. This is a direct optimisation. (**b**) The hierarchical method is used where repetition is applied and only 3 elements are simulated.

　　In Figure 10, we compare the run times and solutions of the two methods. As shown, the hierarchical framework method is capable of finding suitable solutions faster than the direct method. The lower solution time of the hierarchical framework method enables the efficient execution of multiple restarts, which is advantageous when dealing with numerous local optima scattered within the design space. Due to the enormous size of the direct method optimisation problem, solution times are far longer, making multiple restarts infeasible. While direct optimisation may be able to provide better solutions, it is unlikely

that it will find such an optimal solution in a comparable time frame. Table 3 summarises the properties of the optimisation problems in the two scenarios.

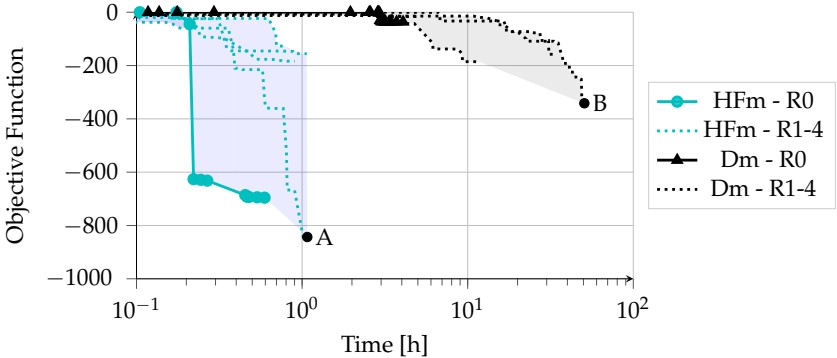

**Figure 10.** Run time and objective function comparison between the hierarchical framework method (HFm) and the direct method (Dm). R represents Restart. The objective function is defined in Equation (3). The solid line shows the results when both methods have the same initial conditions which were a single $1 \times 1$ mm void at the centre of each chamber domain. All other optimisations (shown as dotted lines) have random starting points. Best solution values, **A** = (1.07, −843), and **B** = (50.65, −342).

**Table 3.** Breakdown of different optimisation problems in each method.

| Method | Design Variables | Constraints | Side Constraints | FE Size |
|---|---|---|---|---|
| Hierarchical Framework | 16 | 2 | 32 | 1/3 |
| Direct method | 144 | 2 | 288 | Full |

From the framework above, the possibility arises of using a different HS at each level. By adding a new HS to one level, subsequent levels can leverage its results to uncover diverse design solutions. This occurs as the effects of the HS spread throughout the framework and enable the exploration of new areas of the design space. The capability to add new HSs highlights the scalability of the framework. For instance, the framework can incorporate changes such as recombining units in level 2, or altering the chamber layout in level 3, as shown within the hierarchical framework in Figure 11.

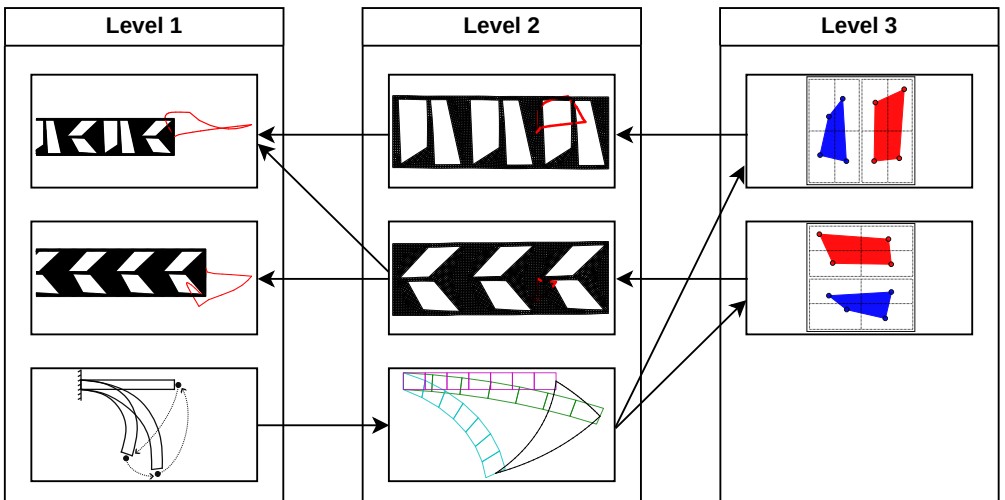

**Figure 11.** An expanded hierarchical framework with examples of additional HSs in level 2 and level 3.

## 5. Conclusions

From the case study presented above, it was shown that using the proposed framework significantly reduced the development time when compared to a direct design approach. The reduced development time also leads to better overall performance, as the reduced computational time can allow for multiple restarts, which aid in more rapidly exploring the design space. The framework allows the designer to make high-level decisions that reduce the computational burden placed on the optimiser by introducing simplifications and creating reduced model representations. Because simplifications inherently reduce the design space, it aids in rapidly finding solutions but can also restrict the final design. The restrictions can be mitigated by using the final solutions of previous simpler designs as a starting point for more direct optimisations. This leverages the reduced design space to obtain a good starting point for a more direct optimisation. While the framework in this study did not incorporate force feedback, it is an essential element for the functionality of soft actuators that interact with the environment. These robots may need to propel themselves forward or manipulate objects, and force feedback is essential to achieving these goals. However, integrating force feedback into the design problem can add additional complexities. Therefore, the next stage in the development of the framework should involve the incorporation of force feedback along with the output displacement to achieve functional soft robots.

**Author Contributions:** Conceptualisation, P.F.L. and M.P.V.; methodology, P.F.L.; software, P.F.L.; formal analysis, P.F.L.; investigation, P.F.L.; resources, M.P.V.; data curation, P.F.L.; writing—original draft preparation, P.F.L.; writing—review and editing, M.P.V.; visualisation, P.F.L.; supervision, M.P.V.; project administration, M.P.V.; funding acquisition, M.P.V. All authors have read and agreed to the published version of the manuscript.

**Funding:** This research was funded in part by the National Research Foundation of South Africa (NRF) grant number 129381.

**Data Availability Statement:** The data presented in this study are available on request from the corresponding author. The data are not publicly available due to this research forming part of a larger study.

**Conflicts of Interest:** The authors declare no conflict of interest.

## Abbreviations

The following abbreviations are used in this manuscript:

| | |
|---|---|
| MDPI | Multidisciplinary Digital Publishing Institute |
| FE | Finite element |
| HS | Hierarchical step |
| SPBA | Soft pneumatic bending actuator |
| PN | Pneumatic network |
| sPN | Slow pneumatic network |
| fPN | Fast pneumatic network |
| MMFD | Modified method of feasible directions |
| HFm | Hierarchical framework method |
| Dm | Direct method |

## Appendix A. Optimisation Loop Details

*Appendix A.1. Non-Linear Solver*

The non-linear solver used was Marc, and the pre- and post-processor used was Mentat [30]. A planar plane strain analysis was performed, using quad4 elements with a maximum size of 0.9 mm. The elements used were Marc element type 118, which are reduced integration and Herman formulation elements. The pressures were applied as edge loads, and self-contact was activated. Convergence properties used in the solver are shown in Table A1.

**Table A1.** Non-linear solver convergence properties.

| Property | Value |
| --- | --- |
| Iterative procedure | Full Newton–Raphson |
| Relative residual force tolerance | 0.1 |

*Appendix A.2. Optimiser*

The design optimiser tool (DOT) from Van der Plaats Research & Development was used as the optimiser [31]. Method 1 of the optimiser, which represents the modified method of feasible directions (MMFD), was used. The optimiser properties are shown in Table A2. The optimiser was not tuned to suit any of the different design methods; rather, the default optimiser parameters were used to eliminate any bias.

**Table A2.** DOT optimiser properties.

| Property | Value |
| --- | --- |
| Absolute Convergence Criteria | $max\left[0.0001 \times ABS(F_0), 1.0 \times 10^{-20}\right]$ |
| Relative convergence criteria | 0.001 |
| Maximum optimisation iterations | 100 |
| Gradient calculation | Forward difference |
| Relative finite difference step | 0.001 |

*Appendix A.3. Function Calls and Evaluation Time*

When running an optimisation, the efficiency of the optimiser can be determined by the number of function calls required to converge to an optimum. In this work, this was not the approach that was taken. The function call is a non-linear FE simulation of an sPN, which has changing geometry. Due to the changing geometry, the problem and, hence, the run time of the simulation change.In the case where the internal voids are small and there is little deformation, the simulation run time can be in the order of 30 s. The run time can increase by a factor 4 or more if the internal voids are larger and there is self contact which occurs. Hence, function calls are not equivalent in this optimisation pipeline, and the optimisation time was rather used as a comparison metric. To ensure that all tests run times could be compared, a single computer was used for all simulations.

*Appendix A.4. Computation Hardware*

The computer that was used for all simulations is detailed in Tables A3 and A4.

**Table A3.** Hardware specifications.

| Processor | Intel(R) Core(TM) i5-6200U CPU @ 2.30 GHz 2.40 GHz |
| --- | --- |
| Installed RAM | 16.0 GB |
| System type | 64-bit operating system, $\times$64 based processor |

**Table A4.** Operating system.

| Operating System | Windows |
| --- | --- |
| Edition | 10 Pro |
| Version | 22H2 |
| OS build | 19,045.2311 |

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
