# Peer review of "A Hierarchical Design Framework for the Design of Soft Robots"

_mca, doi:10.3390/mca28020047_

Round 1

Reviewer 1 Report

This paper describes a hierarchical design framework in developing  environment-specific behaviour for fluid-actuated soft robots. The manuscript is, in general, well written; nevertheless the scientific soundness is not clear. Moreover some reelevant issues are the following:

1. The background and stare of the art must be greatly improved. More relevant references must be added and, more important, the original work must be highlighted with respect to state of the art studies.

2. Equation (1) represents a linear mapping, tough soft robots are known to have nonlinear behaviour; please expand this and put some references.

3. In line110 authors state: "It was found that asymmetrical motion can be achieved by applying two consecutive transformations followed by a return transformation to the original state." Please, clarify how the autors got this result.

4. How the parameters of table 2 were selected.

5. In line 128 one reads: "An objective function is constructed where asymmetrical bending motion is desired." Why asymetrical motion is desired?

6. Figure 6 explains that: "Reduction of a full-length actuator to its simplest representation. All units with unique boundary conditions remain in the reduced representation. Units with similar boundary conditions are condensed into a single unit representative of the group". Why this reduction will lead to the same result as when considerin all the units.?

7. The images are helpful, nonetheless a video animation of the robot motion will be better.

8. Figure 9  represents the paper proposal. This reviewer believes it is straighforward to know that repreating units will lead to a reduced design time, as stated in the abstract and in Figure 10. Why this is one of the main contributions of the paper? How the asembled back is describred o arpplied in this figure?

9. Some typos are found: Table 1: "Cascase to full length", Line 221: "optimier", please perform a careful review.

Reviewer 2 Report

The paper shows a hierarchical framework to design soft robots actuated by pneumatic systems. The paper is well-written and presents a method that can be extensively used by other researchers. The goal of the paper is clearly described and I suggest accepting it, if the following minor comments are addressed.

1) My only comment is about the introduction section: not enough explanation is given about the main core of the paper, namely soft robotics. I would suggest providing a stronger introduction to soft robotic. For example bio-inspiration, 3D printing, and the possibility to create structures with embedded sensors are some pillars that make soft robotcis really appealing . Describing the following (and other soft robotics features) readers would get how important your work (framework to design pneumatic soft robots) is. Here, some papers that can be helpful ( 1)Additive Manufacturing for Bioinspired Structures: Experimental Study to Improve the Multimaterial Adhesion Between Soft and Stiff Materials, 2) Bioinspired three-dimensional-printed helical soft pneumatic actuators and their characterization, 3) One-shot additive manufacturing of robotic finger with embedded sensing and actuation, 4) A 3D printed soft robotic hand with embedded soft sensors for direct transition between hand gestures and improved grasping quality and diversity 5) Soft biomimetic fish robot made of dielectric elastomer actuators

2) another minor comment: if possible can you add in the conclusion which, in your opinion, will be the next step in the design optimization for soft robots.

Round 2

Reviewer 1 Report

All my comments were correctly addressed.